# Mosaic Representation Learning for Self-supervised Visual Pre-training

**Zhaoqing Wang**[1,4]   **Ziyu Chen**[4]   **Yaqian Li**[4]   **Yandong Guo**[4]   **Jun Yu**[3]
**Mingming Gong**[2,*] **Tongliang Liu**[1,*]
[1] Sydney AI Centre, The University of Sydney   [2] The University of Melbourne
[3] University of Science and Technology of China   [4] OPPO Research Institute
zwan6779@uni.sydney.edu.au; mingming.gong@unimelb.edu.au
tongliang.liu@sydney.edu.au

## Abstract

Self-supervised learning has achieved significant success in learning visual representations without the need of manual annotation. To obtain generalizable representations, a meticulously designed data augmentation strategy is one of the most crucial parts. Recently, multi-crop strategies utilizing a set of small crops as positive samples have been shown to learn spatially structured features. However, it overlooks the diversity of contextual backgrounds, which reduces the variance of the input views and degenerates the performance. To address this problem, we propose a mosaic representation learning framework (MosRep), consisting of a new data augmentation strategy that enriches the backgrounds of each small crop and improves the quality of visual representations. Specifically, we randomly sample numbers of small crops from different input images and compose them into a mosaic view, which is equivalent to introducing different background information for each small crop. Additionally, we further jitter the mosaic view to prevent memorizing the spatial locations of each crop. Along with optimization, our MosRep gradually extracts more discriminative features. Extensive experimental results demonstrate that our method improves the performance far greater than the multi-crop strategy on a series of downstream tasks, *e.g.,* $+7.4\%$ and $+4.9\%$ than the multi-crop strategy on ImageNet-1K with 1% label and 10% label, respectively. Code is available at https://github.com/DerrickWang005/MosRep.git.

## 1 Introduction

High-quality representation learning (Bengio et al., 2013) is a fundamental task in machine learning. Tremendous number of visual recognition models have achieved promising performance by learning from large-scale annotated datasets, *e.g.,* ImageNet (Deng et al., 2009) and OpenImage (Kuznetsova et al., 2020). However, a great deal of challenges exist in collecting large-scale datasets with annotations, *e.g.,* label noise (Liu & Tao, 2015; Natarajan et al., 2013; Xia et al., 2019), high cost (Zhu et al., 2019) and privacy concerns (Liang et al., 2020). To address these issues, self-supervised learning (SSL) is proposed to learn generic representations without manual annotation. Recent progress in visual self-supervised learning (Caron et al., 2020; He et al., 2020; Grill et al., 2020; Chen & He, 2021; Bai et al.) shows remarkable potential and achieves comparable results with supervised learning.

Among these SSL methods, a common underlying idea is to extract invariant feature representations from different augmented views of the same input image. Contrastive learning (Dosovitskiy et al., 2015; Wu et al., 2018; Chen et al., 2020a; He et al., 2020; Wang et al., 2022) is one of the most commonly used methods. They define 'positive' and 'negative' pairs and apply the contrastive loss (*i.e.,* InfoNCE (Hénaff et al., 2019)) for optimization, where the 'positive' pairs are pulled close and the 'negative' pairs are pushed away. Another trend of work, such as BYOL (Grill et al., 2020) and Simsiam (Chen & He, 2021), introduces the concept of asymmetry, which is free from designing negatives. They add an extra 'predictor' behind the model and update the parameters using one

---

*Equal contribution

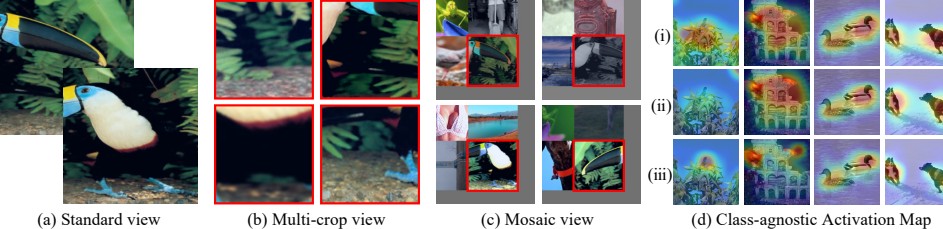

|(a) Standard view|(b) Multi-crop view|(c) Mosaic view|(d) Class-agnostic Activation Map|

Figure 1: (a) The standard view is generated from the input image by applying the strategy used in (He et al., 2020). (b) The multi-crop view is generated from the input image by the multi-crop strategy (Caron et al., 2020). (c) The mosaic view is generated from the input image by our designed augmentation strategy. Although the multi-crop view encourages the "local-to-global" correspondences, it obviously overlooks the diverse background information. In contrast, the mosaic view effectively enriches the background of each crops in the mosaic view, which more facilitates the extraction of discriminative features than the multi-crop strategy. (i), (ii) and (iii) denote the activation maps of MoCo-v2, MoCo-v2 with the multi-crop strategy and MoCo-v2 with our MosRep. Qualitatively, MosRep performs better localization than the other two methods, demonstrating that our method effectively extracts discriminative features and learns high-quality representations.

augmented view, while the feature of another augmented view is used as fixed supervision. Besides, clustering methods (Caron et al., 2018; 2020; Asano et al., 2019; Li et al., 2020) adopt two augmented views of the same image as the prediction and the pseudo cluster label and enforce the consistency between the two views. We present more related works in the appendix.

It is worth noting that a carefully-designed data augmentation strategy is an essential part of the above self-supervised learning frameworks. SimCLR (Tian et al., 2020) and InfoMin (Chen et al., 2020a) empirically investigate the impact of different data augmentations and observe that SSL benefits more from strong data augmentations than supervised learning. After that, SwAV (Caron et al., 2020) proposes the multi-crop strategy, which achieves significant performances on downstream tasks. As shown in Figure 1 (a) and (b), they use two standard resolution crops and sample several small crops that cover the local regions of the input image in order to encourage the "local-to-global" correspondences. However, small crops overlook the diverse backgrounds and decrease the variance, where such views with too many similarities are trivial for learning discriminative features. Intuitively, if we can take into account both the "local-to-global" correspondences and the diverse contextual backgrounds, the quality of learned representations can be further improved.

In this paper, we propose a mosaic representation learning framework (MosRep) consisting of a new data augmentation strategy, which can enrich the contextual background of each small crop and encourage the "local-to-global" correspondences. Specifically, we first sample $M$ (*e.g.,* $M = 4$) small crops of each input image in the current batch. Then, these crops are randomly shuffled and divided into multiple groups. Each group contains $M$ crops and we also ensure that small crops in each group are from different input images. Subsequently, as illustrated in Figure 1 (c), we combine the small crops of the same group into a single view, which terms the mosaic view. Finally, we further jitter the mosaic view in order to prevent the model from memorizing the spatial position of each small crop. In the forward process, the mosaic view is fed into the model for feature extraction, and we adopt the `RoI Align` operator to extract the feature of each crop from the mosaic view and project this feature into an embedding space. To minimize the loss function (*e.g.,* contrastive loss), the model gradually learns to capture more discriminative features (*i.e.,* foreground objects) from the complex backgrounds, improving the quality of visual representations, which is shown in Figure 1 (d). In summary, the main contributions of this paper are as follows:

1. We design a mosaic augmentation strategy that takes into account the diversity of backgrounds and the "local-to-global" correspondences.

2. Based on the proposed mosaic augmentation, we propose a practical and effective SSL framework, MosRep, which benefits the extraction of discriminative features and improves the quality of visual representations.

3. We build our proposed method upon two different SSL frameworks and validate the effectiveness. Experimental results show that our method achieves superior gains compared to the multi-crop strategy on various downstream tasks.

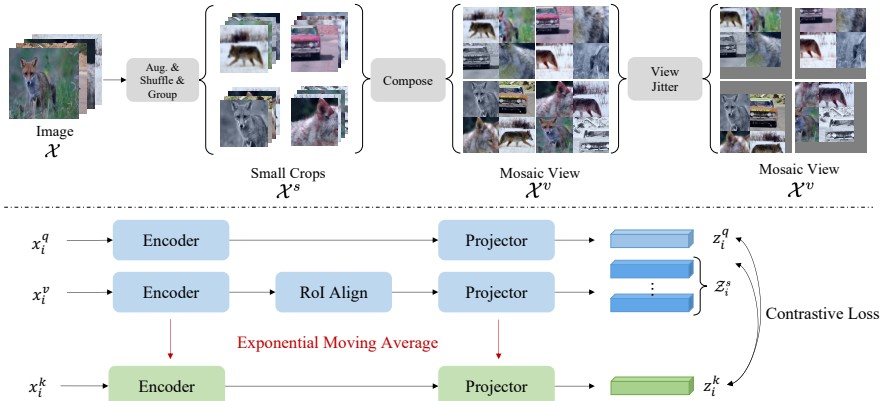

Figure 2: **The framework of MosRep.** The upper figure illustrates the pipeline of the mosaic augmentation strategy. The lower figure illustrates the architecture of the MosRep that is built on the MoCo-v2 (Chen et al., 2020b). Following their training setting, the blue encoder and projector are updated by the SGD optimizer, and the green encoder is updated by the exponential moving average strategy. *Best viewed in color.*

## 2 METHODS

We propose MosRep, a framework to adequately facilitate the learning of discriminative features from large-scale unlabeled datasets. In this section, we first revisit the preliminaries on contrastive learning and the multi-crop strategy. Then we present our mosaic representation learning framework in detail.

### 2.1 PRELIMINARY

**Contrastive learning.** Contrastive learning (Dosovitskiy et al., 2015; Chen et al., 2020a; He et al., 2020) is one of the most popular self-supervised learning frameworks and has achieved great success in recent years. Given a set of images $\mathcal{X}$, the goal of contrastive learning is to learn an embedding space such that the embedding of each image $x_i$ can be distinguished from a set of negatives $\mathcal{N}$. Firstly, two separate augmentations $t^q$ and $t^k$ are sampled from a set of pre-defined augmentations $T$ and applied to obtain two different views of each image, *i.e.*, $x_i^q = t^q(x_i), x_i^k = t^k(x_i)$. Then, these views are fed into an encoder $F(\cdot)$ to extract features $h$ and a projector $g(\cdot)$ to map features $h$ into a embedding space, *i.e.*, $z_i^q = g(F(x_i^q))$, $z_i^k = g(F(x_i^k))$. Finally, a contrastive objective (*e.g.*, InfoNCE) is formulated as:

$$\mathcal{L}_{contrast} = -\frac{1}{N} \sum_{i=1}^{N} \log \frac{\exp(sim(z_i^q, z_i^k)/\tau)}{\exp(sim(z_i^q, z_i^k)/\tau) + \sum_{n^- \in \mathcal{N}} \exp(sim(z_i^q, n^-)/\tau)}, \tag{1}$$

where $n^-$, $\tau$, $N$ and $sim(\cdot)$ denote a negative sample, a temperature parameter, the number of input images and cosine similarity, respectively. Besides, without the need of negatives, BYOL (Grill et al., 2020) and Simsiam (Chen & He, 2021) adopt an extra predictor to map the embedding $z$ to the prediction $p$ and minimize their negative cosine similarity of them, *i.e.*,

$$\mathcal{L}_{cos} = \frac{1}{N} \sum_{i=1}^{N} -sim(p_i^q, sg(z_i^k)) - sim(p_i^k, sg(z_i^q)), \tag{2}$$

where $sg(\cdot)$ denotes the stop-gradient trick that is crucial to avoid model collapsing.

**Multi-crop strategy.** SwAV (Caron et al., 2020) proposes a multi-crop strategy using two high-resolution views that cover large parts of the image and several low-resolution views that cover small parts of the image. In doing so, this strategy maximizes the agreement between a high-resolution (global) view and a low-resolution (local) view of the same image, encouraging the model to learn spatially structured representations.

## 2.2 MOSAIC REPRESENTATION LEARNING

In the multi-crop strategy, the features of low-resolution and high-resolution crops are enforced to remain constant, which encourages the "local-to-global" correspondences. However, the low-resolution crops only cover small parts of the input images, which overlooks the diverse background information and reduces the variance of these crops. Because of this issue, the model is not able to learn sufficient discriminative features, which can cause the degeneration of performance on various downstream tasks.

Motivated by this, we propose a mosaic representation learning framework (MosRep), which consists of a new mosaic augmentation strategy that enriches the backgrounds of each small crop and improves the quality of visual representations. Concretely, given an input image $x_i$, we generate two standard views and $M$ small crops by three separate augmentation operators $t^q$, $t^k$ and $t^s$, *i.e.,*

$$x_i^q = t^q(x_i), \quad x_i^k = t^k(x_i), \quad \mathcal{X}_i^s = t^s(x_i), \qquad t^q, t^k, t^s \sim T, \tag{3}$$

where $x_i^q$ and $x_i^k$ denote two standard view and $\mathcal{X}_i^s$ denotes a set of small crops. As shown in "Small Crops" of Figure 2, we randomly shuffle all small crops from images in a batch and divide them into groups. We set up $M$ crops in each group and ensure that the crops in each group come from different input images. Then, we compose the crops of each group into a single view, termed as the mosaic view $x_i^v$, and record the coordinates $(t, l, b, r)$ of each small crop relative to $x_i^v$, where $t, l, b, r$ indicates the top, left, bottom and right position. This process is formulated as,

$$x_i^v = Compose(\mathcal{M}_i), \quad \mathcal{M}_i = \{x_{ij}^s \mid i \in \mathbf{N}\}_{j=1}^M, \tag{4}$$

where $\mathcal{M}_i$ denotes a group of shuffled small crops, $\mathbf{N}$ denotes all indices of input images and $j$ denotes the index of small crops in a group. That is, $x_{ij}^s$ is the $j$-th crop in the set $\mathcal{X}_i^s$. Since the spatial position of each crop is fixed in the mosaic view $x_i^v$, the model can easily memorize the position, resulting in over-fitting. In order to tackle this dilemma, we conduct the view jitter operation on the mosaic view. We first sample offsets of the mosaic view from a beta distribution $\beta(\alpha, \alpha)$ with two identical parameters $\alpha$, *i.e.,*

$$\Delta x = \theta \cdot u, \quad \Delta y = \theta \cdot v, \qquad u, v \sim \beta(\alpha, \alpha), \tag{5}$$

where $\theta$, $\Delta x$ and $\Delta y$ denote the upper bound of jitter range and the offsets of the mosaic view, respectively. We set $\alpha < 1$, which indicates a U-shaped distribution. In this way the mosaic view is more likely to be jittered in a relatively wider range with larger offsets. Meanwhile, we update the coordinates of each small crop with these offsets, which is calculated as,

$$(t', l', b', r') = (t + \Delta y, l + \Delta x, b + \Delta y, r + \Delta x). \tag{6}$$

In doing so, the mosaic view effectively enriches the background information of each small crop and facilitates the extraction of discriminative features, improving the quality of learned representations. Subsequently, we present the forward propagation of our proposed framework in the lower part of Figure 2. Given two standard views $x_i^q, x_i^k$ and a mosaic view $x_i^v$, an encoder $F(\cdot)$ is used to extract the feature $h$ of them, *i.e.,*

$$h_i^q = F(x_i^q), \quad h_i^k = F(x_i^k), \quad h_i^v = F(x_i^v), \tag{7}$$

where $h_i^q$, $h_i^k$ and $h_i^v$ denote the feature of two standard views and a mosaic view, respectively. It is worth noting that, by resorting to the above-mentioned coordinates, we adopt a `RoI Align` operator to extract the feature of each small crop in the mosaic view. According to the index $i$ of the input images, we rearrange the features of these small crops to correspond to their positive samples, and use $\mathcal{H}_i^s$ to denote a set of features of small crops in $i$-th mosaic view. After that, all features are mapped into an embedding space by a projector $g(\cdot)$, *i.e.,*

$$z_i^q = g(h_i^q), \quad z_i^k = g(h_i^k), \quad \mathcal{Z}_i^s = g(\mathcal{H}_i^s), \tag{8}$$

where $z_i^q$, $z_i^k$ are the embedding of two standard views, and $\mathcal{Z}_i^s$ denotes a set of embeddings of small crops. Finally, we define the positive and negative pairs of all embeddings based on the index $i$ of the input images and calculate the contrastive loss, *i.e.,*

$$\begin{aligned}
\mathcal{L}_{contrast} = &- \frac{1}{N} \sum_{i=1}^{N} \log \frac{\exp(sim(z_i^q, z_i^k)/\tau)}{\exp(sim(z_i^q, z_i^k)/\tau) + \sum_{n_j \in \mathcal{N}} \exp(sim(z_i^q, n_j)/\tau)} \\
&- \frac{1}{N} \sum_{i=1}^{N} \frac{1}{M} \sum_{j=1}^{M} \log \frac{\exp(sim(z_{ij}^s, z_i^k)/\tau)}{\sum_{n^- \in \mathcal{Z}^k \bigcup \mathcal{N}} \exp(sim(z_{ij}^s, n^-)/\tau)},
\end{aligned} \tag{9}$$

where $z_{ij}^s$ denotes the $j$-th embedding in the set $\mathcal{Z}_i^s$, $\mathcal{Z}^k$ denotes a set of embeddings of the key view and $n^-$ denotes negative samples in the set $\mathcal{Z}^k$ and $\mathcal{N}$. The first component in Eq. 9 is the standard contrastive loss used in (Chen et al., 2020b). And we design the second component in Eq. 9 to explicitly pull the small crops and its corresponding key view closer and push other irrelevance away. With optimization, the model gradually learns to extract more discriminative features, thus improving the quality of learned representations. Besides, we also build our proposed method upon the BYOL (Grill et al., 2020) framework by minimizing the negative cosine similarity between small crops and the key view:

$$Loss = -\frac{1}{B}\sum_{j\in B}\sum_{k\in B'}\frac{\exp(p_{c_{jk}}^{i2t}/\tau^T)}{\sum_{k\in B'}\exp(p_{c_{jk}}^{i2t})}\cdot\log\frac{\exp(p_{s_{jk}}^{i2t}/\tau^S)}{\sum_{k\in B'}\exp(p_{s_{jk}}^{i2t})} \tag{10}$$

$$-\frac{1}{B}\sum_{j\in B}\sum_{k\in B'}\frac{\exp(p_{c_{jk}}^{t2i}/\tau^T)}{\sum_{k\in B'}\exp(p_{c_{jk}}^{t2i})}\cdot\log\frac{\exp(p_{s_{jk}}^{i2i}/\tau^S)}{\sum_{k\in B'}\exp(p_{s_{jk}}^{i2i})}, \tag{11}$$

where $B$ and $B'$ denote the batch of each rank and all ranks, $p_c^{i2t}$ and $p_c^{t2i}$ denote the image-to-text similarity and the text-to-image similarity of the CLIP model, $p_s^{i2i}$ and $p_s^{i2t}$ denote the image-to-image similarity and the image-to-text similarity between the small model and CLIP model, and $\tau^T$ and $\tau^S$ denote the temperature of the CLIP model and small model.

## 3 EXPERIMENTAL RESULTS

### 3.1 PRE-TRAINING SETTINGS

**Datasets**   We perform self-supervised pre-training on two datasets, one middle-scale and another large-scale: 1) 100-category ImageNet (IN-100) (Tian et al., 2019), a subset of IN-1K dataset containing $\sim$125k images; and 2) 1000-category ImageNet (IN-1K) (Deng et al., 2009), the standard ImageNet training set containing $\sim$1.25M images.

**Architectures**   Following the commonly-used setting in recent unsupervised methods (He et al., 2020; Chen et al., 2020a; Tian et al., 2019), we adopt the ResNet-50 (He et al., 2016) model as our encoder. To study the flexibility of our proposed method, we build on two different frameworks, MoCo-v2 (Chen et al., 2020b) and BYOL (Grill et al., 2020).

**Data Augmentation**   During pre-training, we adopt the data augmentation used in MoCo-v2 (He et al., 2020) and BYOL (Grill et al., 2020) for the standard view. As for the mosaic view, we generate $M = 4$ small crops with $112 \times 112$ input size, and other data augmentations are the same with MoCo-v2 and BYOL.

**Optimization**   Our hyperparameters closely follow these of the adopted self-supervised learning methods, MoCo-v2 (He et al., 2020) and BYOL (Grill et al., 2020). As for the MoCo version, we pre-train the network on IN-100 and IN-1K for 400 and 200 epochs, respectively. SGD (Loshchilov & Hutter, 2016) optimizer with a cosine learning rate scheduler and $lr_{base} = 0.3$ is adopted, with a mini-batch size of 256 on 8 NVIDIA V100 GPU. We utilize a negative queue of 16,384 for IN-100, and 65,536 for IN-1K. The weight decay is 0.0001 and SGD momentum is 0.9.

As for the BYOL (Grill et al., 2020) version, we pre-train the network on IN-100 for 400 epochs, and on IN-1K for 200 epochs. We adopt the SGD optimizer with $lr_{base} = 0.7$, where the learning rate is linearly scaled with the batch size as $lr = lr_{base} \times batch/256$. The batch size is set as 512 for IN-1OO and 2048 for IN-1K. The weight decay is 0.000001 and the SGD momentum is 0.9. For the momentum encoder, the momentum value starts from 0.99 and is increased to 1, following (Grill et al., 2020). We use batch normalization (Ioffe & Szegedy, 2015) synchronized across devices.

### 3.2 EVALUATION ON IMAGENET

To evaluate the effectiveness of our proposed method, we pre-train the model on IN-100 and IN-1K, respectively, and conduct a series of downstream tasks, including linear probing, nearest neighbor, semi-supervised learning, and transfer learning.

Table 1: **Evaluation on ImageNet linear probing.** We adopt a ResNet-50 (He et al., 2016) as our backbone, and pre-train it on middle-scale and large-scale datasets, *i.e.,* ImageNet-100 and ImageNet-1K (Deng et al., 2009) for 400 and 200 epochs, respectively. We build our method on two widely-used frameworks, MoCo-v2 (Chen et al., 2020b) and BYOL (Grill et al., 2020) to evaluate the universality. To thoroughly verify the effectiveness, we set the standard two-crop version as a weak baseline and a multi-crop version as a strong baseline. We report the top 1 accuracy of linear probing and $k$-NN on ImageNet-100 and ImageNet-1K, respectively. $*$ denotes the strong baseline.

| Method | ImageNet-100 | | | ImageNet-1K | | |
|---|---|---|---|---|---|---|
| | Linear | 1-NN | 5-NN | Linear | 1-NN | 5-NN |
| Supervised | 85.8 | 85.3 | 94.5 | 76.5 | 74.9 | 90.2 |
| SimCLR | - | - | - | 68.3 | - | - |
| SwAV | - | - | - | 69.1 | - | - |
| SimSiam | - | - | - | 70.0 | - | - |
| MSF | - | - | - | 71.4 | - | - |
| MoCo-v2 | 80.9 | 75.0 | 90.9 | 67.7 | 55.7 | 78.6 |
| MoCo-v2$*$ | 83.8 | 75.7 | 91.9 | 69.8 | 56.0 | 80.0 |
| MosRep (Ours) | **85.7** | **78.2** | **92.6** | **72.3** | **61.7** | **81.9** |
| BYOL | 82.3 | 79.5 | 92.1 | 72.4 | 66.1 | 85.0 |
| BYOL$*$ | 83.2 | 78.9 | 91.8 | 74.7 | 69.6 | 86.6 |
| MosRep (Ours) | **84.7** | **80.9** | **92.4** | **76.2** | **70.4** | **87.4** |

**Linear Probing** To evaluate the quality of visual representations, we conduct linear classification on IN-100 and IN-1K, following a standard protocol used in (Chen & He, 2021). We freeze the deep features and train a supervised classifier for 90 epochs with a batch size of 1024. We adopt the SGD optimizer with $lr_{base} = 0.1$, where the learning rate is linearly scaled with the batch size as $lr = lr_{base} \times batch/256$. We use standard supervised ImageNet augmentations (He et al., 2020) during training. For a thorough comparison, we set two baselines: 1) a weak baseline with two standard crops. 2) a strong baseline with the multi-crop strategy.

Firstly, we build our MosRep upon the MoCo-v2 baseline. The key difference between the two frameworks is the use of mosaic representation learning. As illustrated in Table 1, MosRep surpasses the weak baseline by an obvious margin of 4.8% at linear probing on IN-100. Similarly, MosRep achieves a significant gain of 4.6% over the weak baseline at linear probing on IN-1K. More importantly, MosRep shows a considerable improvement over the strong baseline on both IN-100 and IN-1K datasets, which demonstrates that the performance gain is not simply from more small crops. For each small crop, the mosaic view effectively introduce diverse background information. We argue that, along with the optimization, the model needs to extract more discriminative features from the diverse background to maximize the similarity of positive pairs and minimize the similarity of negative pairs, which can improve the quality of learned representations. In the BYOL experiments, we observe similar phenomenon as the MoCo-v2 ones.

**Nearest Neighbor** We further evaluate the representations of the pre-trained models by the nearest neighbor classifier. Following (Koohpayegani et al., 2021), we adopt the center crop operation with $256 \times 256$ on the training and test sets, and calculate $l2$ normalized embeddings by forwarding through the backbone. We report Top-1 accuracy on 1-NN and 5-NN classifiers. For a thorough comparison, we set two baselines: 1) a weak baseline with two standard crops. 2) a strong baseline with the multi-crop strategy.

Table 1 shows the results of nearest neighbor on the IN-100 and IN-1K datasets, respectively. MosRep achieves more consistent improvements over the weak and strong baselines on the 1-NN and 5-NN classifiers. Significant gains of 2.5% and 5.7% are achieved on the IN-100 and IN-1K datasets when we build our method on the MoCo-v2 framework. Besides, we also build on BYOL and obtain similar improvements of 2.0% and 0.8% on the IN-100 and IN-1K, respectively.

**Semi-supervised Learning** Following the semi-supervised settings (Chen et al., 2020a; Grill et al., 2020; Hénaff et al., 2019), we evaluate the pre-trained models on the task of classification with

Table 2: **Evaluation on ImageNet-1K semi-supervised training.** We pre-train a ResNet-50 (He et al., 2016) on ImageNet-1K for 200 epochs, and conduct semi-supervised training on 1% and 10% labelled sets. We build our method on two widely-used frameworks, MoCo-v2 (Chen et al., 2020b) and BYOL (Grill et al., 2020) to evaluate the universality. To fully evaluate the effectiveness, we set the standard two-crop version as a weak baseline and the multi-crop version as a strong baseline. Top-1 and Top-5 accuracy are used as benchmark metrics. $*$ denotes the strong baseline.

| Method | ImageNet-1K (1%) | | ImageNet-1K (10%) | |
|---|---|---|---|---|
| | Top-1 | Top-5 | Top-1 | Top-5 |
| SimCLR | 44.8 | - | 59.5 | - |
| SwAV | 52.5 | - | 67.2 | - |
| SimSiam | 46.8 | - | 62.4 | - |
| MoCo-v2 | 43.5 | 70.9 | 58.9 | 82.8 |
| MoCo-v2$*$ | 45.4 | 73.9 | 60.8 | 84.9 |
| MosRep (Ours) | **52.8** | **78.7** | **65.7** | **87.5** |
| BYOL | 54.1 | 78.9 | 66.9 | 87.5 |
| BYOL$*$ | 58.4 | 81.3 | 68.8 | 88.9 |
| MosRep (Ours) | **60.0** | **82.7** | **70.2** | **89.5** |

Table 3: **Evaluation on Linear probing transfer learning evaluation.** We pre-train a ResNet-50 (He et al., 2016) on ImageNet-1K for 200 epochs, and conduct linear probing on 8 classification datasets. We build our method on two widely-used frameworks, MoCo-v2 (Chen et al., 2020b) and BYOL (Grill et al., 2020) to evaluate the universality. $*$ denotes the strong baseline.

| Method | CIFAR 10 | CIFAR 100 | STL 10 | Food 101 | Flower 102 | DTD | Pets | Cars |
|---|---|---|---|---|---|---|---|---|
| Supervised | 90.7 | 73.5 | 97.0 | 73.0 | 85.9 | 67.2 | 91.9 | 47.9 |
| MoCo-v2 | 90.7 | 72.7 | 95.6 | 71.4 | 82.6 | 68.1 | 81.7 | 42.4 |
| MoCo-v2$*$ | 90.6 | 72.4 | 96.8 | 72.0 | 78.3 | 69.8 | 80.5 | 35.9 |
| MosRep (Ours) | **91.5** | **74.5** | **97.1** | **74.3** | **84.5** | **71.3** | **84.5** | **44.8** |
| BYOL | 92.4 | 77.1 | 96.8 | 72.9 | 87.8 | 70.3 | 88.3 | 56.5 |
| BYOL$*$ | 93.5 | **78.8** | 97.5 | 77.0 | **92.0** | 71.6 | 91.0 | **65.0** |
| MosRep (Ours) | **93.7** | 78.2 | **97.8** | **77.3** | 90.9 | **72.5** | **91.2** | 63.0 |

limited ImageNet labels. The sizes of annotations are reduced to 1% and 10% on the IN-1K (Deng et al., 2009) training dataset, respectively. We freeze the backbone and train a single linear classifier. We utilize the features from the top of the backbone, which are normalized to have unit $l2$ norm and then scaled and shifted to have zero mean and unit variance for each dimension. The linear classifier is trained with SGD optimizer (lr=0.01, epochs=40, batch size=256, weight decay=0.0005, and momentum=0.9). The learning rate is multiplied by 0.1 at 15 and 30 epochs. We use the standard setting of ImageNet supervised learning [1] during training.

Table 2 illustrates the comparisons of our approach against two widely-used frameworks (He et al., 2020; Grill et al., 2020) and their strong variants. Impressively, when we build our proposed MosRep on the MoCo-v2, we achieves considerable improvements over the strong baseline with both 1% and 10% labels, *e.g.,* 7.4% Top-1 accuracy on 1% subset and 4.9% Top-1 accuracy on 10% subset. After that, we build on the BYOL and still significantly outperforms the strong baseline by 1.4% and 0.6% at Top-1 accuracy with 1% and 10% labels, respectively.

### 3.3 EVALUATION ON TRANSFER LEARNING

To further evaluate the quality of learned representations, we conduct a series of transfer learning, including linear probing on various classification datasets, COCO object detection (Lin et al., 2014), COCO instance segmentation (Lin et al., 2014) and cityscapes instance segmentation (Cordts et al., 2016).

---

[1]https://github.com/pytorch/examples/blob/master/imagenet/main.py

Table 4: **Evaluation on COCO object detection and instance segmentation.** We adopt a ResNet-50 (He et al., 2016) as our backbone, and pre-train it on ImageNet-100 and ImageNet-1K (Deng et al., 2009) for 400 and 200 epochs, respectively. Each pretrained model is transferred to the Mask R-CNN R50-FPN model, subsequently finetuned on COCO 2017 (Lin et al., 2014) train set and evaluated on COCO 2017 val set. Averaging precisions on bounding-boxes ($AP^{bb}$) and masks ($AP^{mk}$) are used as benchmark metrics. $*$ denotes the strong baseline.

| Method | ImageNet-100 | | ImageNet-1K | |
|---|---|---|---|---|
| | $AP^{bb}$ | $AP^{mk}$ | $AP^{bb}$ | $AP^{mk}$ |
| Supervised | 37.2 | 33.7 | 38.9 | 35.4 |
| MoCo-v2 | 37.6 | 34.1 | 39.8 | 36.0 |
| MoCo-v2$^*$ | 38.2 | 34.6 | 40.4 | 36.6 |
| MosRep (Ours) | **39.0** | **35.3** | **40.6** | **36.6** |
| BYOL | 38.3 | 34.7 | 40.4 | 36.7 |
| BYOL$^*$ | 38.5 | 35.0 | 40.9 | 37.1 |
| MosRep (Ours) | **38.6** | **35.2** | **41.1** | **37.2** |

**Linear Probing**   Following the procedure adopted in (Chen et al., 2020a; Grill et al., 2020), we perform the linear probing task on the following datasets: CIFAR-10 (Krizhevsky et al., 2009), CIFAR-100 (Krizhevsky et al., 2009), STL-10 (Coates et al., 2011), Food-101 (Bossard et al., 2014), Flower-102 (Nilsback & Zisserman, 2008), DTD (Cimpoi et al., 2014), Pets (Parkhi et al., 2012) and Cars (Krause et al., 2013).

First, we build on the MoCo-v2 framework and compare it with the weak and strong baselines. As shown in the Table 3, it is obvious that our proposed MosRep outperforms both baselines on all eight datasets, and achieves better transfer performances than the supervised pre-trained model on CIFAR-10, CIFAR-100, STL-10, Food-101 and DTD, which demonstrates that our method can effectively improve the generalization. Besides, we also conduct experiments on the BYOL framework. Although BYOL is a state-of-the-art self-supervised learning framework, which achieves excellent transfer performances on eight datasets, our approach can improve the generalization performance on most datasets, even outperforming the IN-1K supervised model across the board.

**COCO Object Detection & Instance Segmentation**   Following the procedure outlined in (Chen et al., 2020b; He et al., 2020; Chen & He, 2021), we fine-tune a Mask-RCNN (R50-FPN) (He et al., 2017; Lin et al., 2017; Wu et al., 2019) under $1\times$ scheduler and add new Batch Normalization layers before the FPN parameters. We report Average Precision on bounding-boxes ($AP^{bb}$) and Average Precision on masks ($AP^{mk}$). Table 4 shows the comparison of our proposed methods and both baselines. Our MosRep can consistently increase the ability of detection and segmentation on the MoCo-v2 and BYOL frameworks.

**Cityscapes Instance Segmentation**   Following the setting utilized in ReSim Xiao et al. (2021), we fine-tune a Mask-RCNN (R50-FPN) He et al. (2017); Lin et al. (2017); Wu et al. (2019) and add Batch Normalization layers before the FPN, and synchronize all Batch Normalization during training. Table 5 illustrates the comparisons of MosRep versus the supervised pre-training counterparts and the weak and strong baselines.

### 3.4   ABLATIONS

In this section, we study the effect of some design choices in the MosRep, including the jittering operation, $\alpha$ in the beta distribution and the upper bound $\theta$ of jitter range. We built our method upon the MoCo-v2 and pre-train it on IN-100 for 400 epochs. We report the Top-1 accuracy on IN-100 linear probing.

**Effect of $\alpha$ in the Beta distribution**   We use $\alpha$ values from the set $\{0.3, 0.5, 1.0, 2.0\}$ and set $\alpha = 0.5$ for the above experiments. A smaller $\alpha$ means a greater likelihood of having a large range of jittering and a larger $\alpha$ means the opposite. As illustrated in Figure 3, MosRep with the

Table 5: **Evaluation on Cityscapes instance segmentation.** We adopt a ResNet-50 (He et al., 2016) as our backbone, and pre-train it on ImageNet-100 and ImageNet-1K (Deng et al., 2009) for 400 and 200 epochs, respectively. To evaluate the effectiveness, we set the standard two-crop version as a weak baseline and the multi-crop version as a strong baseline. Each pretrained model is transferred to Mask R-CNN R50-FPN model, subsequently finetuned on Cityscapes (Cordts et al., 2016) train set and evaluated on Cityscapes val set. Averaging precision on masks ($AP^{mk}$) is used as benchmark metric. All results are the average of three trials. $^*$ denotes the strong baseline.

| Method | ImageNet-100 | | ImageNet-1K | |
|---|---|---|---|---|
| | $AP^{mk}$ | $AP^{mk}_{50}$ | $AP^{mk}$ | $AP^{mk}_{50}$ |
| Supervised | 29.4 | 54.6 | 32.9 | 59.6 |
| MoCo-v2 | 32.2 | 58.3 | 33.7 | 60.5 |
| MoCo-v2$^*$ | 32.8 | 59.9 | 34.0 | 61.0 |
| MosRep (Ours) | **33.4** | **61.0** | **34.7** | **64.0** |
| BYOL | 28.2 | 56.5 | 33.4 | 62.4 |
| BYOL$^*$ | 29.5 | **58.0** | 33.9 | 63.5 |
| MosRep (Ours) | **30.0** | 57.3 | **34.2** | **63.8** |

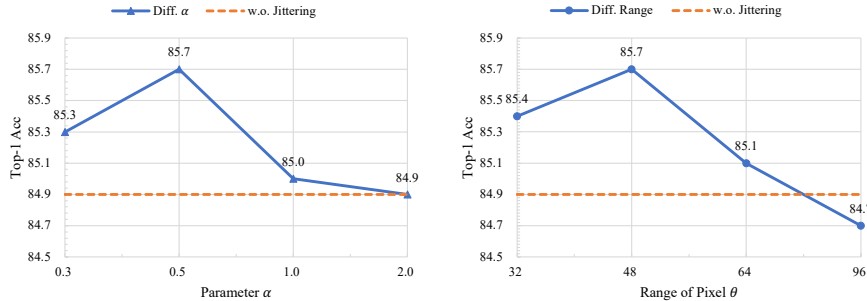

Figure 3: Built on the MoCo-v2, we train the MosRep on the IN-100 for 400 epochs, and conduct the linear prob on the IN-100. The orange dashed line denotes the MosRep without the jitter operation. jittering operation achieves better transfer performance than ones without the jittering operation, which demonstrates that this operation can increase the variance of the mosaic view to prevent the model from memorizing the spatial location of each small crop.

**The Jitter Range** We further study the effect of different jitter ranges. As shown in the right part of Figure 3, with an increasing jittering range, MosRep achieves better Top-1 accuracy on the linear probing. However, when the jittering range is too large, more false positive samples with no overlapping area between positive pairs are likely to be observed, which results in the degradation of the quality of visual representations.

## 4 CONCLUSIONS

In this paper, we design a mosaic data augmentation strategy that focuses on small parts of input images and adequately enriches the background information. Through this strategy, we develop a simple and effective mosaic representation learning framework for self-supervised visual pre-training. Our method can be flexibly combined with other self-supervised learning frameworks, *e.g.,* MoCo-v2 and BYOL. Extensive experimental results demonstrate that our method consistently boosts the performance on a series of downstream tasks, including linear probing, nearest neighbor, semi-supervised learning, transfer learning, and etc. We hope this work could inspire future research on view designing, considering its significant role in self-supervised learning.

The main limitation is that although we achieved promising results on various image-level downstream tasks, we do not observe noticeable gains on pixel-level downstream tasks, especially when pre-trained on large-scale datasets. This phenomenon implies that image-level self-supervised learning is sub-optimal for pixel-level downstream tasks. We will study this problem in our future research.

## 5 ACKNOWLEDGE

The authors would like to thank the anonymous reviewers and the meta-reviewer for their constructive feedback and encouraging comments on this work. Zhaoqing Wang was supported by OPPO Research Institute. Jun Yu is sponsored by Natural Science Foundation of China (62276242), CAAI-Huawei MindSpore Open Fund (CAAIXSJLJJ-2021-016B, CAAIXSJLJJ-2022-001A), Anhui Province Key Research and Development Program (202104a05020007), USTC-IAT Application Sci. & Tech. Achievement Cultivation Program (JL06521001Y). Mingming Gong was supported by ARC DE210101624. Tongliang Liu was partially supported by Australian Research Council Projects IC-190100031, LP-220100527, DP-220102121, and FT-220100318.

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

# A    RELATED WORK

Self-supervised learning methods have been extensively studied to close the gap with supervised learning. These methods can be mainly categorized into several aspects, including contrastive-based, consistency-based, clustering-based and generative-based methods.

## A.1    CONTRASTIVE LEARNING

contrastive self-supervised learning has emerged as a promising approach to unsupervised visual representation learning. Each image is treated as an individual class in an instance discrimination setting, whose core idea is to pull the positive pairs together and push the negative pairs away in the embedding space. As studied in many previous works (Chen et al., 2020b; He et al., 2020; Chen et al., 2020a; Hu et al., 2021; Li et al., 2022), it is extremely essential to construct high-quality positive and negative pairs to achieve higher transfer performance. Specifically, InstDisc (Wu et al., 2018) increases the size of negative pairs by built a memory bank that stores pre-computed embeddings, thus improving the performance. Following this line, MoCo (He et al., 2020) utilizes a momentum update mechanism to update a large queue of negatives for contrastive learning. This momentum encoder greatly improves the quality and consistency of negative pairs and achieves remarkable performance compared to previous works. SimCLR (Chen et al., 2020a) further improves in a straightforward way that directly adopts negative samples in the current batch with much bigger batch size. Besides, researchers carefully construct a rich family of data augmentations on cropped images, significantly boosting classification accuracy. MoCo-v2 Chen et al. (2020b) also improve the performance than MoCo (He et al., 2020) by using the same data augmentation and MLP layer design.

## A.2    CONSISTENCY REGULARIZATION

Different from other contrastive-based methods, BYOL (Grill et al., 2020) trains an online network to predict the output of the target network, with the latter slowly updated with momentum. The authors assume that the additional predictor to the online network and the momentum encoder are essential to avoid collapsed solutions without negatives. SimSiam (Chen & He, 2021) further explores simple Siamese networks that can learn transferable representations without negative samples, large batches and momentum encoders. The role of the stop gradient is emphasized in preventing collapse. BarlowTwins (Zbontar et al., 2021) avoids collapse by measuring the cross-correlation matrix between the outputs of two identical networks fed with distorted versions of a sample, and making it as close to the identity matrix as possible. This causes the embedding vectors of distorted versions of a sample to be similar, while minimizing the redundancy between the components of these vectors.

## A.3    CLUSTERING

These methods, such as DeepCluster (Caron et al., 2018), SWAV (Caron et al., 2020), SELA Ma et al. (2019), perform contrastive-like comparisons without the requirement to compute all pairwise distances. Specifically, these methods cluster the data simultaneously while enforcing consistency between cluster assignments produced for different distortions of the same image, instead of directly comparing features in contrastive learning. Clustering methods are also prone to collapse, e.g., empty clusters in k-means and avoiding them relies on careful implementation details. Online clustering methods like SWAV can be trained with large and small batches but require storing features when the number of clusters is much larger than the batch size.

## A.4    GENERATIVE & MASK IMAGE MODELING

The generative methods typically adopt auto-encoders (Kingma et al., 2019), and adversarial learning to train an unsupervised representation. Usually, they focused on the pixel-wise information of images to distinguish images from different classes. For instance, BigBiGAN (Donahue & Simonyan, 2019) adopted BiGAN to capture the relationship between latent semantic representations and the input images. Motivated by the great success of BERT, masked image modeling (MIM) (Xie et al., 2022; Bao et al., 2021; He et al., 2022) becomes a new trend in self-supervised visual pre-training, which randomly masks parts of images and reconstructs them based on the corrupted image. ViT (Dosovitskiy et al., 2021) attempts to adopt masked patch prediction for self-supervised learning.

Table 6: **Evaluation on ImageNet linear probing.** By plugging in DINO Caron et al. (2021), we adopt a ViT-S/16 (Dosovitskiy et al., 2021) as our backbone, and pre-train it on middle-scale and large-scale datasets, *i.e.,* ImageNet-100 and ImageNet-1K (Deng et al., 2009) for 200 epochs. To thoroughly verify the effectiveness, we set the standard two-crop version as a weak baseline and a multi-crop version as a strong baseline. We report the top 1 accuracy of linear probing and 1-NN on ImageNet-100 and ImageNet-1K, respectively. * denotes the strong baseline.

| Method | ImageNet-100 | | ImageNet-1K | |
|---|---|---|---|---|
| | Linear | 1-NN | Linear | 1-NN |
| DINO | 49.3 | 41.2 | 62.6 | 53.1 |
| DINO* | 66.7 | 56.1 | 68.5 | 56.1 |
| MosRep (Ours) | **70.9** | **60.0** | **70.0** | **63.1** |

Table 7: **Ablation study of adding simple background noise.** By plugging in MoCo-v2 Chen et al. (2020b), we adopt a ResNet-50 (He et al., 2016) as our backbone, and pre-train it on ImageNet-100 for 400 epochs. We report the top 1 accuracy of linear probing and 1-NN on ImageNet-100.

| Method | Linear | 1-NN |
|---|---|---|
| MoCo-v2 `w/` Multi-crop | 83.8 | 75.7 |
| MoCo-v2 `w/` Multi-crop (Noise) | 83.6 | 75.1 |
| MosRep (Ours) | **85.7** | **78.2** |

BEiT (Bao et al., 2021) predicts the discrete tokens of masked token resorting to an off-the-shelf discrete VAE. Instead of discretizing the visual information, MAE (He et al., 2022) and SimMIM (Xie et al., 2022) propose to directly predict the pixel-level value as the reconstruction target.

Table 8: **Ablation study of different input size of small crops.** By plugging in MoCo-v2 Chen et al. (2020b), we adopt a ResNet-50 (He et al., 2016) as our backbone, and pre-train it on ImageNet-100 for 400 epochs. We report the top 1 accuracy of linear probing and 1-NN on ImageNet-100.

| Method | Input size | Linear | 1-NN |
|---|---|---|---|
| MosRep | $64 \times 64$ | 82.9 | 74.2 |
| MosRep | $80 \times 80$ | 84.9 | 77.2 |
| MosRep | $96 \times 96$ | 85.5 | 77.8 |
| MosRep | $112 \times 112$ | **85.7** | **78.2** |

# B  ADDITIONAL EVALUATION ON IMAGENET

We build our method on DINO Caron et al. (2021). For a fair comparison, we set a standard two-crop version as a weak baseline and a multi-crop version as a strong baseline. We perform self-supervised pre-training and linear probing on both ImageNet-100 and ImageNet-1K. Following Caron et al. (2021), we adopt ViT-S/16 as the encoder and pre-train for 200 epochs. The batch size is 1024. We report the top1 accuracy for linear probing and 1-NN. As illustrated in Table 8, MosRep surpasses both baselines by clear margins on linear probing and 1-NN.

# C  ADDITIONAL ABLATION STUDY

**Simple background noise.** We conduct an ablation study that adds simple background noise (e.g., gaussian) in the small crops. Experimental results in the following table 1 show that adding simple background noise cannot improve the performance of linear probing. We argue that although simple

background noise can enrich the background information, it cannot effectively reduce the redundant mutual information and is easy to optimize.

**Different input sizes of small crops.** We vary the input size of small crops from $64 \times 64$ to $112 \times 112$ and pre-train a ResNet-50 on ImageNet-100 for 400 epochs. As illustrated in Table 1, experimental results show that decreasing the input size of small crops causes performance degradation on linear probing and 1-NN. Considering the limited resource, we do not further increase the input size of small crops.

## D  PSEUDO-CODES OF MOSAIC AUGMENTATION STRATEGY

**Algorithm 1:** Mosaic augmentation strategy

```
1 # x:  the input images batch with shape of (B, 4, C, H, W)
2 # a:  the parameter used in Beta distribution
3 x = x.reshape(B * 4, C, H, W)
4 x_mos = shuffle(x, dim=0)
5 x_mos = x_mos.reshape(B, 2, 2, C, H, W)
6 x_mos = x_mos.permute(0, 3, 1, 4, 2, 5)
7 x_mos = x_mos.reshape(B, C, 2 * H, 2 * W)
8 s = beta(a, a)
9 x_mos_jit = jitter(x, s)
```

