# OpenReview forum: "Mosaic Representation Learning for Self-supervised Visual Pre-training"
_ICLR.cc/2023/Conference — ICLR 2023 notable top 25%_

### Official Review · Reviewer_hAGo · 2022-10-21

**Confidence:** 3
**Correctness:** 3
**Technical Novelty And Significance:** 3
**Empirical Novelty And Significance:** Not applicable
**Recommendation:** 8

**Clarity, Quality, Novelty And Reproducibility:**

Please see my strengths and weaknesses section. Overall, I think the clarity and reproducibility can be improved especially around the method section.

**Strength And Weaknesses:**

Strength:
The technique is simple and is well-motivated. The multi-crop strategy leads to low-resolution negative sample, which can be of limited use if the background in the image are textured and include a lot of visual detail. By grabbing higher resolution patch and place it into the view, the author have achieve a more challenging background for the model to learn from.
The improvement over the multi-crop technique is appreciable. The difference ranges from 0.5-1.5 point, which is not large, but appreciable.
The experiments are comprehensive, and I find enough details are given to be reproducible. The comparison is fair, as the author retrain the model used for comparison. It may have been useful to actually tuned hyperparameter per augmentation strategy as well. As written, it seems like the hyperparameters were pre-selected and applied across board.

Weakness:
I find the clarity of writing to be the main issue here. It would be much simpler to explain the augmentation algorithm as well as the representation learning methodology using a diagram. Using equations is prone to error, and also harder to understand. As written, there are some errors (index i is overloaded: i-th mosaic, and also i-th image in eq 8). This makes a text that is already hard to understand even harder.
No code was promised, following the above point, without a clearly explained method, it is hard to reimplement the work here. I believe there is no promise of code release here, so I think it can be challenging for other researcher to try to reimplement this exactly.
Lastly, the paper can be stronger if the ablation study is used to justify technical choices. For example, one main motivation for this work is to have a more complex background. It may be interesting for the author to try to vary the resolution of the background patch in the mosaic, and see if there is any change in model performance. As presented, it seems more like a tuning exercise for alpha and theta.

**Summary Of The Paper:**

This paper proposes a simple modification of multi-view augmentation for use with representation learning, as well as an additional training framework to go with it. The main idea is to create a richer set of background with higher resolution than available with the multi-crop strategy. Because the augmented data is mosaiced, the author apply ROI-align to extract only relevant ROI of the feature map before transforming it to the embedding space. The contrastive loss is applied to the appropriate part of the embedding from the standard view of the input image as well as from its mosaiced version. Experiments are given and demonstrate 0.5-1.5% improvement over the multi-crop technique.

**Summary Of The Review:**

I think there is enough contribution here in the paper. However, the clarity and reproducibility concern limited the impact to the research community. I find that to address this will require a major revision of the text and figure, but a code release may alleviate this somewhat.

---

> ### Author Response · Authors · 2022-11-18
> **Responce to reviewer hAGo**
>
> **Q1.** Improving the clarity of the proposed method.
>
> **A1.** To promote clarity, we add the pseudo-codes of the mosaic augmentation strategy in our appendix.
>
> ---
>
> **Q2.** The mistake of Eq. (8).
>
> **A2.** Thanks for pointing out this mistake. We have fixed it in the next version, and we have carefully checked our implementation and found it correct.
>
> ---
>
> **Q3.** Ablation study to vary the resolution of the background patch in the mosaic view.
>
> **A3.** Thanks for the comments. We vary the input size of small crops from $64\times64$ to $112\times112$ and pre-train a ResNet-50 on ImageNet-100 for 400 epochs. As illustrated in Table 1, experimental results show that decreasing the input size of small crops causes performance degradations on linear probing and 1-NN. Considering the limited resources, we do not further increase the input size of small crops.
>
> **Table 1. Ablation study of different input sizes of small crops.**
> | Method | Input size | Top@1 | 1-NN |
> | :----: | :----: | :----: | :----: |
> | MoCo-v2 *w/*  MosRep | $64\times64$ | 82.9 | 74.2 |
> | MoCo-v2 *w/*  MosRep | $80\times80$ | 84.9 | 77.2 |
> | MoCo-v2 *w/*  MosRep | $96\times96$ | 85.5 | 77.8 |
> | MoCo-v2 *w/*  MosRep | $112\times112$ | **85.7** | **78.2** |

---

> > ### Author Response · Authors · 2022-11-21
> > **Responce to code release**
> >
> > We plan to release our code after the final decision.

---

### Official Review · Reviewer_rkci · 2022-10-24

**Confidence:** 3
**Correctness:** 3
**Technical Novelty And Significance:** 2
**Empirical Novelty And Significance:** 3
**Recommendation:** 5

**Clarity, Quality, Novelty And Reproducibility:**

The paper is well written and easy un follow. It's novelty is limited; it build on top of the multi-crop strategy from SwAV with a few interesting improvements. Enough implementation details are provided for at least partial reproducibility.

**Strength And Weaknesses:**

-- Strengths:
- The proposed method is reasonable and results show imprevements on performance from it
- The approach was tested on multiple benchmark datasets

-- Weaknesses:
- Novelty is limited
- Performance evaluation is focused on camparing to standard BYOL and MoCo-v2 after adding the proposed augmentation strategy. However, it does not outperform more recent SSL approaches.

**Summary Of The Paper:**

Authors proposed a mosaic augmentation strategy for better representation learning in a self-supervised learning setup. The proposed MosRep strategy was applied to both MoCo-v2 and BYOL frameworks improving it's performance by a considerable margin on ImageNet-100 and ImageNet-1k datasets. The approach was further evaluated for transfer learning for both object detection and instance segmentation using MS COCO and Cityscapes with positive results.

**Summary Of The Review:**

The proposed mosaic augmentation strategy and MosRep framework appears to be a good augmentation strategy to learn good representation in MoCo-v2 and BYOL contrastive learning frameworks. The novelty is limited, and there is no comparison with most recent SOTA SSL approaches but the results are promising for multiple tasks.

---

> ### Author Response · Authors · 2022-11-18
> **Responce to reviewer rkci**
>
> **Q1.** Novelty is limited.
>
> **A1.** We thank the reviewer for reviewing the paper. However, we respectfully disagree on the comments. We have addressed the low variance problem of positive pairs in the multi-crop strategy. The low variance problem of positive pairs is widely existed and under explored, addressing which will greatly contribute to the community.
>
> Regarding to novelty, we think there are several aspects, i.e., the concept novelty, technical novelty, experimental novelty, and so on. We believe our paper is novel in the aforementioned aspects. For example, for concept novelty, we are the first to address the low variance problem of positive pairs in the multi-crop strategy. We have clearly stated the research problem and the strong motivation. For technical novelty, our mosaic augmentation idea is novel and hasn't been explored in self-supervised learning, which is specifically designed to address the low variance problem of positive pairs in the multi-crop strategy. For experimental novelty, we have provided results better than the state-of-the-art methods in a wide range, e.g., we have conducted extensive experiments on a serials of downstream tasks with better results. We are confused about the comment. Could you please provide reasons why you believe novelty is limited?
>
> ---
>
> **Q2.** Not outperform more recent SSL approaches.
>
> **A2.** Thanks for the comments. Our proposed method is a plug-and-play framework, which is flexible to build on many self-supervised learning (SSL) methods. We have built our method on a more recent method, DINO [1], and conducted experiments on IN-100 and IN-1K. For a fair comparison, we adopt a standard two-crop version as the weak baseline and a multi-crop version as the strong one. Experimental results demonstrate that our proposed MosRep consistently surpasses two baselines.
>
> **Table 1. Results of linear probing and 1-NN on ImageNet-100.**
> | Method | Backbone | Epoch | Crop Size | Top@1 | 1-NN |
> | :----: | :----: | :----: |:----: | :----: | :----: |
> | DINO *w/* Two-crop | ViT-S/16 | 200 | 2 $\times$ 224 | 49.3 | 41.2 |
> | DINO *w/*  Multi-crop | ViT-S/16 | 200 | 2 $\times$ 224 + 4 $\times$ 96 | 66.7 | 56.1 |
> | DINO *w/*  MosRep | ViT-S/16 | 200 | 2 $\times$ 224 + 4 $\times$ 96 | **70.9** | **60.0** |
>
> **Table 2. Results of linear probing and 1-NN on ImageNet-1K.**
> | Method | Backbone | Epoch | Crop Size | Top@1 | 1-NN |
> | :----: | :----: | :----: | :----: | :----: | :----: |
> | DINO *w/* Two-crop | ViT-S/16 | 200 | 2 $\times$ 224 | 62.6 | 53.1 |
> | DINO *w/*  Multi-crop | ViT-S/16 | 200 | 2 $\times$ 224 + 4 $\times$ 96 | 68.5 | 59.6 |
> | DINO *w/*  MosRep | ViT-S/16 | 200 | 2 $\times$ 224 + 4 $\times$ 96 | **70.6** | **63.1** |
>
> We hope we have addressed your concerns. As you haven’t pointed out any specific SSL approach. If you still have concerns, we would appreciate it if you could point out the other SSL methods you think we need to compare.
>
> [1] Caron M, Touvron H, Misra I, et al. Emerging properties in self-supervised vision transformers[C]//Proceedings of the IEEE/CVF International Conference on Computer Vision. 2021: 9650-9660.

---

> ### Author Response · Authors · 2022-11-28
> **Do you still have concerns?**
>
> Dear Reviewer rkci,
>
> Do you have any remaining concerns? We believe your participation in the rolling discussion will help improve the paper.
>
> Best wishes,
> Authors

---

> ### Author Response · Authors · 2022-12-01
> **Interaction**
>
> Dear Reviewer rkci,
>
> Thanks for helping review the submission. We are approaching the end of the rolling discussion phase. We are looking forward to your feedback.
>
> Best wishes, Authors

---

### Official Review · Reviewer_ed1B · 2022-10-25

**Confidence:** 4
**Correctness:** 4
**Technical Novelty And Significance:** 2
**Empirical Novelty And Significance:** 2
**Recommendation:** 8

**Clarity, Quality, Novelty And Reproducibility:**

Clarify: The paper is mostly clearly written.

Quality: Empirical results are comprehensive and show definite improvements over multi-crop.

Reproducibility: It is a simple and therefore easy to reproduce method.

Novelty: The idea is straightforward, and so I may not consider it particularly "novel". However, a simple, useful idea is valuable to the community.

**Strength And Weaknesses:**

Strengths:

1. The method is simple, although possibly details of implementations could be a little complicated (e.g. to ensure that features are re-arranged to correlate small and large crop views).

2. Strong empirical results on various tasks such as classification and linear transfer.

Weaknesses:

There is a very limited understanding of the idea other than "diverse backgrounds". What exactly improves the features? Did the authors try ablation such as simply adding background noise to the views?

I'm curious how this approach could be applied to vision transformers, since those architectures are increasingly more common.

**Summary Of The Paper:**

This paper proposes an alternative to multi-crop data augmentation. Multi-crop augmentation has been shown to provide consistent improvements in representation learning. This is achieved by using 2 "large crop" views that capture "global" information of the image, and a number of much smaller crops that capture the "local" information.

The authors argue that naive small crops tend to lead to lowered mutual information between views (although they do not use the term "mutual information"), thereby leading to suboptimal performance. The authors argue that by creating synthetic backgrounds for the small crops, mutual information is reduced, thereby leading to stronger (more discriminative) features.

A comprehensive set of experiments shows the benefits of their approach on classification and transfer learning tasks.

**Summary Of The Review:**

The paper has a very simple idea supported by a lot of empirical evidence. Hence I lean towards acceptance. However, there is a very limited understanding of why this approach works.

---

> ### Author Response · Authors · 2022-11-18
> **Response to reviewer ed1B**
>
> **Q1.** The understanding of the idea other than "diverse backgrounds". What exactly improves the features? Did the authors try ablation such as simply adding background noise to the views?
>
> **A1.** Thanks for this valuable comment. By applying the multi-crop strategy, we can obtain two large-resolution crops and several small-resolution crops. The large crop usually contains the global information of foreground objects, while the small crop contains the local information of objects. The alignment of a positive pair of large and small crops encourages the model to capture the “local-to-global” information. However, these small crops easily share too much overlapped information with large crops, which is trivial for learning discriminative representations. Our designed mosaic augmentation strategy composes several small crops into a large crop and enriches each small crop's background information, effectively reducing redundant information and keeping the minimal sufficient part. In doing so, the model needs to extract more discriminative features to minimize the optimization term.
>
> Besides, following your suggestion, we conduct an ablation study that adds simple background noise (e.g., gaussian) in the small crops. Experimental results in the following table 1 show that adding simple background noise cannot improve the performance of linear probing. We argue that although simple background noise can enrich the background information, it cannot effectively reduce the redundant mutual information and is easy to optimize.
>
> **Table 1. Ablation study of adding background noise.**
> | Method | Backbone | Epoch | Top@1 | 1-NN |
> | :---- | :----: | :----: | :----: |:----: |
> | MoCo-v2 *w/* Multi-crop | R50 | 400 | 83.8 | 75.7 |
> | MoCo-v2 *w/*  Multi-crop (Noise) | R50 | 400 | 83.6 | 75.1 |
> | MoCo-v2 *w/*  MosRep | R50 | 400 | **85.7** | **78.2** |
>
> ---
>
> **Q2.** The effectiveness on Vision Transformers (ViT).
>
> **A2.** Great question! We are also curious about the effectiveness of our proposed method built on vision transformers. We build our method on DINO [1]. For a fair comparison, we set a standard two-crop version as a weak baseline and a multi-crop version as a strong baseline. We perform self-supervised pre-training and linear probing on both ImageNet-100 and ImageNet-1K . Following [1], we adopt ViT-S/16 as the encoder and pre-train for 200 epochs. The batch size is 1024. We report the top1 accuracy for linear probing and 1-NN. As illustrated in Table 2 and 3, MosRep surpasses both baselines by clear margins on linear probing and 1-NN.
>
> **Table 2. Results of linear probing and 1-NN on ImageNet-100.**
> | Method | Backbone | Epoch | Crop Size | Top@1 | 1-NN |
> | :----: | :----: | :----: |:----: | :----: | :----: |
> | DINO *w/* Two-crop | ViT-S/16 | 200 | 2 $\times$ 224 | 49.3 | 41.2 |
> | DINO *w/*  Multi-crop | ViT-S/16 | 200 | 2 $\times$ 224 + 4 $\times$ 96 | 66.7 | 56.1 |
> | DINO *w/*  MosRep | ViT-S/16 | 200 | 2 $\times$ 224 + 4 $\times$ 96 | **70.9** | **60.0** |
>
> **Table 3. Results of linear probing and 1-NN on ImageNet-1K.**
> | Method | Backbone | Epoch | Crop Size | Top@1 | 1-NN |
> | :----: | :----: | :----: | :----: | :----: | :----: |
> | DINO *w/* Two-crop | ViT-S/16 | 200 | 2 $\times$ 224 | 62.6 | 53.1 |
> | DINO *w/*  Multi-crop | ViT-S/16 | 200 | 2 $\times$ 224 + 4 $\times$ 96 | 68.5 | 56.1 |
> | DINO *w/*  MosRep | ViT-S/16 | 200 | 2 $\times$ 224 + 4 $\times$ 96 | **70.6** | **63.1** |
>
> [1] Caron M, Touvron H, Misra I, et al. Emerging properties in self-supervised vision transformers[C]//Proceedings of the IEEE/CVF International Conference on Computer Vision. 2021: 9650-9660.

---

### Public Comment · ~whisper_ai1 · 2023-04-03
**RoI Align operator and code release**

#

Thank you for your interesting work. And congratulations on the acceptance of your article by ICLR2023!

But I still have some questions, I'm looking forward to your answers

---

In the article you mention that “It is worth noting that, by resorting to the above-mentioned coordinates, we adopt a RoI Align operator to extract the feature of each small crop in the mosaic view”

**Q**：You project this feature into an embedding space. Is the $z_{ij}^{s}(dim) = z_{i}^{k}(dim)$ ? It would be helpful if you could elaborate on how the projections are made.

---

In the article you mention that “In order to tackle this dilemma, we conduct the view jitter operation on the mosaic view”

**Q:** What is the padding type due to the offset of the coordinates? You use ROI Align operator in the later operation, does such an alignment introduce noise (e.g.  a view containing other negative samples or containing padding )? It would be helpful if you could elaborate on how the ROI Align operator works.

---

Congratulations again on the acceptance of your article by ICLR2023! **Hopefully the code and logs will be made public soon.**

---

### Decision · Program_Chairs · 2023-01-20

**Decision:**

Accept: notable-top-25%

**Justification For Why Not Higher Score:**

It's possible to boost the paper to oral but the main reason of not higher is the scope of the paper. Although the paper has impact on the broad ML community, a better data augmentation strategy is still an improvement to an existing module of the ML pipeline.

**Justification For Why Not Lower Score:**

The findings of this paper is not only highly interesting to the self-supervised learning community, but to the broad community where data augmentation is needed.

**Metareview: Summary, Strengths And Weaknesses:**

The paper proposed a novel mosaic version of multi-view augmentation for representation learning by using 2 “large crop” views that capture “global” information of the image, and much smaller crops that capture the “local” information. The mutual information between different views is effectively reduced by creating synthetic backgrounds for the small crops. Experiments demonstrated considerable improvement (0.5~1.5%) over the multi-crop technique on classification and transfer learning tasks.

Two reviewers (R# ed1B and R# hAGo) gave very positive reviews. They both feel that the method is simple, well-motivated and effective, and strong empirical results were obtained. They expressed the weakness of clarity of writing (the authors added the pseudo-code of the mosaic augmentation strategy in the appendix), the limited understanding of the idea e.g. what exactly improves the features and what about simply adding background noise to the views (the authors addressed it with additional ablation studies and convincing results). Both reviewers were content with the rebuttal.

Reviewer #rkci was a bit negative with the concern that the novelty is limited. Unfortunately R #rkci did not respond to the author's comments nor AC’s request. The AC does not agree with the comment that the novelty is limited because “it builds on top of the multi-crop strategy from SwAV with a few interesting improvements.” The authors addressed the R #rkci’s question of “not outperforming more recent SSL approaches” with very convincing numbers on DINO.

The AC feels that the paper obtained strong results by a novel augmentation method with good motivation and justification. Both the methodology and results have broad impact to the broad ML community.


**Note From Pc:**

if the above contains the word "oral" or "spotlight" please see: "oral" presentation means -> notable-top-5% and "spotlight" means -> notable-top-25%. As stated in our emails, we are disassociating presentation type from AC recommendations